# Comparing the Productivity of the Latest Models of Li-Ion Battery and Petrol Chainsaws in a Conifer Clear-Cut Site

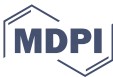

Andrea Laschi [1], Francesco Neri [2,*], Elena Marra [2], Fabio Fabiano [2], Niccolò Frassinelli [2], Enrico Marchi [2], Riccardo Paoloni [2] and Cristiano Foderi [2]

1 Dipartimento di Scienze Agrarie, Alimentari e Forestali—SAAF, Università Degli Studi di Palermo, Viale Delle Scienze Ed. 4., 90128 Palermo, Italy
2 Dipartimento di Scienze e Tecnologie Agrarie, Alimentari, Ambientali e Forestali—DAGRI, Università Degli Studi di Firenze, Via San Bonaventura 13, 50145 Firenze, Italy
* Correspondence: francesco.neri@unifi.it

**Abstract:** The recent technological development of batteries has allowed the production of powerful tools that are also used in forestry operations. For this reason, this study aimed to compare the performance of two latest chainsaw models in a conifer clear-cut. The examined chainsaws, the battery-powered Stihl MSA 300 and the petrol-powered MS 261 C-M, have comparable power and weight. The overall working times were recorded and then gross, net and felling/processing productivities were quantified. Our findings reported that in a working day (7.4 h gross time), each chainsaw felled and processed an average of 20 trees of 0.64 m$^3$ with an average diameter at breast height of 28 cm. Considering the net productivity, no statistically significant differences were recorded between the two chainsaws. Investigating the productivities on operations conducted using the chainsaws, the battery chainsaw showed a lower average tree-processing productivity than that recorded for the petrol chainsaw. On average, the battery duration was 0.88 h, while the petrol tank duration was 0.97 h. Our study shows that battery chainsaws have great potential in forestry operations; however, considering the actual need for 8 charges per standard workday, optimizing battery management is an important objective for future work.

**Keywords:** productivity investigation; conifer clear-cut; battery; petrol; chainsaw





## 1. Introduction

In recent years, the market for battery-powered tools in the green maintenance and forestry sector has been continuously increasing, reaching 51% of sales worldwide, and sales of tools with this power supply are expected to increase further [1]. Recent developments in battery technology have enabled the production of powerful, compact and lightweight manual tools even in the forestry sector [2,3], which has high demands on the power and reliability of the tools. In the last few decades, the density energy of rechargeable batteries has increased ten times, passing from Ni-MH technology to modern Li-ion cells [1]. Nowadays, the most widely used batteries are lithium-ion ones, which have good energy and power density, durability, safety and a significantly reduced weight compared to older battery types [4]. The growing interest in battery-powered tools has also affected the industrial sector of chainsaws. Chainsaw use is very common in professional forestry [5], rescue operations, and arboriculture but also in household and gardening activities. Petrol chainsaws are normally used in forestry operations, while battery and electric models are usually chosen for green maintenance or pruning operations [6]. However, each working sector needs the appropriate chainsaw technical specifications (weight, power and handling). High power and reliability are especially required in the forestry and rescue fields [7], as the machine is used at full power for many hours.

With increasing attention on workers' well-being and environmental issues [8], leading international chainsaw brands have begun to develop battery-powered models, as they can

create less harmful working conditions for operators [9–13]. In fact, due to the absence of the piston-crack system present in internal combustion chainsaws, and consequently of the forward and backward movement of the piston [14], battery-powered chainsaws generate lower noise emission levels and fewer vibrations [3,6]. This results in increased safety and protection for the forest operator, with a reduced risk of developing occupational health problems, such as hand–arm vibration syndrome (HAVs), vibration-induced white fingers (VWF), and hearing loss [7,15–18].

As regards environmental aspects, battery tools do not have an endothermic engine, so they do not produce exhaust fumes, and modern Li-ion batteries have a recycling efficiency of 97% $w/w$ (percent weight in weight) of the battery's valuable active materials [19,20]. Comparably no exhaust gases on site, easy maintenance and the absence of cables are other advantages of the battery chainsaws [21]. Despite these positive aspects, battery-powered chainsaws still show some drawbacks, such as insufficient battery capacity and the risk of overheating of the battery pack during hard work, with possible consequences on productivity [1]. Previous studies by Spinelli et al. [22], Colantoni et al. [23] and Engelbrecht et al. [24] showed that in gardening and arboriculture activities, where branches and small trees often need to be cut, the battery chainsaw can be a viable alternative.

To date, studies carried out on battery-powered chainsaws have been concerned mainly with the evaluation of the operator's exposure to noise and vibrations under controlled conditions [3,6,14], but few studies have investigated the productivity and the reliability of battery-powered chainsaws in regular forestry operations [7,25]. In terms of performance, cutting efficiency is affected by several factors, such as the wood density, i.e., tree species, degree of wood contamination [26,27], moisture content, chain filing and type [28]. In particular, the higher the wood density, the higher the cutting force requirements. The first findings of the studies mentioned above showed that battery-powered chainsaws also have great potential in forestry operations due to their reliability and productivity, but the limiting factor is the numerous charges required to complete the entire working day (8 h) in the forest.

The objective of this research was to compare the productivity ($m^3h^{-1}$) of two latest models of Stihl chainsaws with similar technical specifications (MS 261 C-M petrol-powered and MSA 300 battery-powered), in a conifer clear-cut site in Central Italy under standard working conditions (8 h a day). Our research hypothesis was that the felling and processing productivity is not significantly different for the two chainsaws.

## 2. Materials and Methods

### 2.1. Study Site

The study site was in the Montebello Forest (province of Forli-Cesena, Emilia Romagna, Italy; 44°09′51.7″ N, 11°79′57.1″ E). The study was conducted in a stand stocked with a 50-year-old Scots pine (*Pinus sylvestris* L.) plantation and, within it, two adjacent sections were selected for testing. The silvicultural treatment applied in each section was clear-cutting. Clear-cutting was chosen to fully compare the performance of the two chainsaws in tree felling and processing operations, excluding the delays often occurring in thinning operations due to the presence of trees to be released. Tree characteristics and site conditions were previously checked for similarity to achieve a valid comparison between the two chainsaw models. The average terrain slope was 5%–10%. Tree diameters at breast height (DBH) were analysed to ensure uniformity, while tree heights were homogenous, given the same age of the trees in the stand.

The first section (S1) covered 0.214 ha, with a tree density of 729 stems/ha (155 trees cut) and mean tree basal area of 48 $m^2ha^{-1}$ and a mean DBH of 28.5 cm. The second section (S2) covered 0.101 ha, with a tree density of 891 stems/ha (90 trees cut), a mean tree basal area of 51 $m^2ha^{-1}$ and a mean DBH of 26.8 cm. Trees felled and processed were previously marked by forest technicians of the Montebello Forest service; in each section, six trees were spared as reserve trees, as required by local legislation. In both sections, a few trees

were dead and/or leaned towards others. They were removed before the beginning of the study to avoid differences in operational conditions.

The clear-cut was performed by two forest operators with more than ten years of experience and who were qualified as expert forest instructors by a government public institution (Tuscany Region). Operators cut daily, alternating the use of the battery and petrol chainsaw, and they worked daily in a different portion of each section, moving from S1 to S2. This operational scheme was created to allow operators to work at a safe distance from each other every day and to use both battery and petrol chainsaws in each section.

### 2.2. Description of the Equipment Used

To compare their operational performance in regular forestry operations, two latest models of chainsaws, manufactured by Stihl S.p.A., were tested in the conifer clear-cut (tree felling and processing). Table 1 shows their technical specifications. In detail, the MSA 300 (Li–ion battery-powered, indicated as BAT) and the MS 261 C-M (petrol-powered, indicated as MIX) were chosen for the study. BAT was selected as having the best performance among the professional battery chainsaws available on the market (as advertised at the beginning of 2022 by the Stihl manufacturer). MIX was chosen for comparison since, according to the manufacturer, it has similar technical specifications (weight and power). Furthermore, MIX has the technical specifications (weight and power) of the most versatile and most appreciated chainsaws available on the market for medium-sized tree utilization. The two compared models were chosen from the same manufacturer in order to use the same cutting device with the same technical specifications.

**Table 1.** Technical specifications of the two chainsaws.

| | Stihl MSA 300 | Stihl MS 261 C-M |
|---|---|---|
| **Technical Specification** | BAT | MIX |
| **Power** | 3.0 kW | 4.1 kW |
| **Saw-bar length** | 40 cm | 40 cm |
| **Chain type** | Half-chisel | Half-chisel |
| **Chain pitch** | 0.325″ (0.8255 cm) | 0.325″ (0.8255 cm) |
| **Drive-link thickness** | 1.3 mm | 1.3 mm |
| **Number of drive links** | 67 | 67 |
| **Fuel supply** | Electricity (battery) | Mixed (gasoline + oil) |
| **Battery/Fuel type** | AP500S | Stihl MotoMix |
| **Maximum chain speed (ISO 11681)** | 30 m·s$^{-1}$ | 25.6 m·s$^{-1}$ |
| **Total weight \*** | 7.7 kg | 6.9 kg |

\* Including saw bar, chain and battery or fuel and chain lubricant.

For the comparison, the productivity of the two chainsaws (m$^3$h$^{-1}$) was analysed and calculated by collecting the working times and measuring the wood volume harvested. The total volume harvested was calculated by measuring log diameters and the height of whole trees.

### 2.3. Time Study

During the clear-cut, four researchers (two per operator) collected data on working times, DBH, whole tree height and delays. Working times were recorded using centesimal (1 min = 100 min) stopwatches. Work time was divided into 15 working phases [29], reported separately for each operator. The operations that did not require the use of a chainsaw were split into two sub-phases and the productive chainsaw time (Using Chainsaw, indicated as "UC") was separated from the productive time consumed using other equipment (Not Using Chainsaw, indicated as "NUC"). Briefly, the studied workday and the tree felling and processing cycle were analysed as follows (Figure 1):

- *General preparation*: includes the walk from the track to the worksite at the beginning of the workday, the preparation of tools, and the operational checks before starting harvesting operations.
- *Approach and tree study:* includes all the preparatory activities made with and without the chainsaw before starting to cut the notch (e.g., the cleaning of the worksite from undergrowth, the tree evaluation and the choice of the felling technique).
- *Felling cut*, "UC": includes all the felling operations made with the chainsaw, from the first cut of the notch to the beginning of tree falling.
- *Felling cut*, "NUC": includes all the felling operations made without the chainsaw (e.g., the check of the correct fall direction, the use of wedges, etc.).
- *Tree falling*, "UC": includes the time spent using the chainsaw to allow the tree to fall after the end of the felling cut. Normally it is related to resolving situations where a tree becomes caught in another.
- *Tree falling*, "NUC": includes the time spent to make the tree fall on the ground after the end of the felling cut.
- *Stump cleaning*, "UC": includes the cuts to trim both the stump and the bottom of the first trunk. It ends when the operator hooks up the meter to measure the trunks.
- *Delimbing*, "UC": includes all the time spent by the operator in active delimbing with the chainsaw.
- *Delimbing*, "NUC": includes those operations where the operator has interrupted the delimbing to move the cut branches along the trunk or to turn the trunk to complete the branches' removal from the part below.
- *Cross-cutting*, "UC": includes only the time spent by the operator in cross-cutting.
- *Cross-cutting*, "NUC": includes the time spent to measure the trunks' length and eventual time spent without using the chainsaw.
- *Refuelling:* starts when the battery or petrol is used up during the operation until the chainsaw is ready again to continue.
- *Chain sharpening:* includes the time spent for sharpening.
- *Rest time.*
- *Delays:* problems and drawbacks.

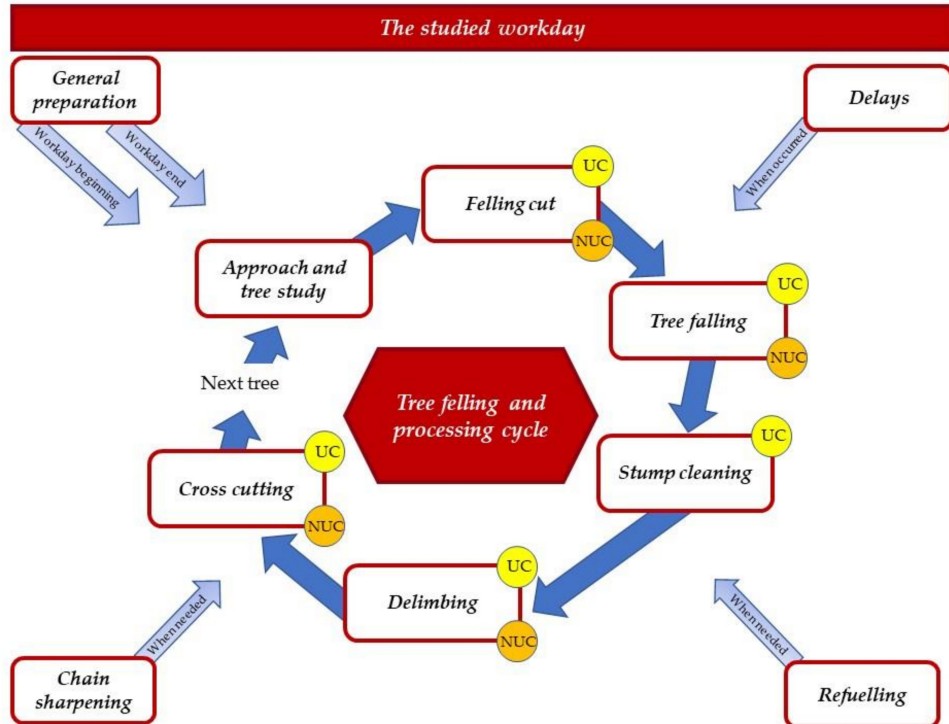

**Figure 1.** Time study flowchart. UC = using chainsaw, NUC = not using chainsaw.

The wood volume harvested during each work cycle was combined with time consumption to calculate productivity [30].

### 2.4. Productivity Calculation

Productivity was investigated in regular forestry operations using battery- and petrol-powered chainsaws. In the study, the following productivities were investigated for the two chainsaw models: (1) gross productivity; (2) net productivity; (3) productivity in felling and processing; and (4) UC productivity. The research hypothesis was that BAT and MIX had a similar average productivity ($m^3h^{-1}$) in different working phases. To evaluate the potential influence of the chainsaw model on productivity, UC and NUC chainsaw working times were collected separately, since in some phases the operator used the chainsaw but in other phases, other tools were needed. For example, during felling, the operator cut the notch (UC) and later used wedges or other instruments (NUC) to facilitate the tree falling. Table 2 reports the working phases included in all the productivities investigated.

**Table 2.** Working phases and productivities investigated, showing where each working phase is included or excluded.

| Working Phase (Using Chainsaw "UC" or Not Using Chainsaw "NUC") | Productivity | | | | |
|---|---|---|---|---|---|
| | Gross | Net | Felling/Processing | UC | NUC |
| General preparation | x | | | | |
| Approach and tree study | x | | | | |
| Felling cut—UC | x | x | x | x | |
| Felling cut—NUC | x | x | x | | x |
| Tree falling—UC | x | x | x | x | |
| Tree falling—NUC | x | x | x | | x |
| Stump cleaning—UC | x | x | x | x | |
| Delimbing—UC | x | x | x | x | |
| Delimbing—NUC | x | x | x | | x |
| Cross-cutting—UC | x | x | x | x | |
| Cross-cutting—NUC | x | x | x | | x |
| Refuelling | x | x | | | |
| Chain sharpening | x | | | | |
| Rest time | x | | | | |
| Delays | x | | | | |

Gross productivity, as the ratio between the wood volume harvested (by BAT and MIX) and the total working hours per day, includes all the above-mentioned working phases. Net productivity, as the ratio between the wood volume harvested and the effective working hours per day (considering using (UC) and not using (NUC) chainsaw phases plus the refuelling time) is described in Table 2. As well as the productivities mentioned above, the felling and processing productivities were calculated considering the wood volume daily harvested divided by the total time spent for felling and processing less the refuelling time (Table 2). To compare BAT and MIX productivity only in the productive phases, the wood volume harvested was divided by the UC working time.

### 2.5. Statistical Analysis

The productivity dataset calculated for BAT and MIX was analysed. Normality was checked with the Shapiro test and homoscedasticity of variance with the Bartlett test. Then, a parametric approach was used. In terms of productivities, the differences between BAT and MIX were verified by implementing a one-way ANOVA and Tukey's honestly significant differences (HSD) a post-hoc test. Moreover, the influence of tree size (diameter and height) on productivity for each chainsaw model and for each operator was tested with the same approach. Multi-way ANOVA was applied to the productivity dataset to test the effects caused by the chainsaw model, operators and possible interactions between chainsaw models and operators. BAT and MIX UC productivities were compared using

firstly a non-parametric statistical approach, due to a non-normal distribution data, and then with a Kruskal–Wallis rank sum test since data were homoscedastic.

## 3. Results

In the clear-cut area, a total of 245 trees were felled and processed: 125 trees with BAT and 120 with MIX. On average, BAT felled and processed 12.78 m$^3$ per day, while MIX achieved 13.36 m$^3$. The stand analysed had an average diameter of 28 cm (range between 16 cm and 44 cm), an average height of 21 m and an average tree wood volume of 0.64 m$^3$. No statistically significant differences were recorded in the diameters cut by the two chainsaws and by the two operators (Table 3).

**Table 3.** Minimum, maximum, mean and SD values for the diameter size of the cut trees. Different superscript letters highlight a significant difference (*p*-value < 0.05).

| Chainsaw Model | Operator | N. | Minimum | Maximum | Mean | SD |
|:---:|:---:|:---:|:---:|:---:|:---:|:---:|
|  | 1 | 54 | 16.00 | 37.00 | 27.04 [a] | 5.85 |
| **BAT** | 2 | 71 | 18.00 | 44.00 | 27.70 [a] | 4.79 |
|  | Overall | 125 | 16.00 | 44.00 | 27.42 [a] | 5.26 |
|  | 1 | 59 | 19.00 | 39.00 | 28.20 [a] | 4.20 |
| **MIX** | 2 | 61 | 16.00 | 42.00 | 28.40 [a] | 5.80 |
|  | Overall | 120 | 16.00 | 42.00 | 28.35 [a] | 5.05 |

To fell and process all the trees (125 by BAT and 120 by MIX), 51 battery charges (i.e., 8.5 per day on average) and 45 refuels (i.e., 7.5 per day on average) were needed. On average, 2.45 trees were felled and processed per battery charge and 2.67 trees per full tank of petrol. Considering the gross working time, the battery charge lasted 0.88 h and the full tank 0.97 h, on average. As for the UC working time, the battery charge lasted 0.40 h and the full tank 0.50 h. During the six working days, the overall gross working time was 44.87 h for BAT and 43.57 h for MIX (i.e., 7.4 working hours per day). BAT and MIX performed 20.53 h and 22.53 h of UC working time, respectively. This corresponds to 48% of the gross working time for BAT and 52% for MIX. Comparing the productivities of the two operators, significant differences were found in gross, net, felling and processing and NUC productivities (Table 4). No differences were recorded between them in UC productivity.

**Table 4.** Analysis of variance (*p*-values) of the effects of chainsaw model (BAT-battery MSA 300; MIX-petrol MS 261 C-M), operator (1 or 2) and their interactions on the productivities (ANOVA multiple comparison test for gross, net, felling and processing productivity and not using chainsaw "NUC"; Kruskal–Wallis rank sum test for using chainsaw "UC" productivity).

| Source of Variance for Productivity | Gross | Net | Felling/Processing | "UC" | "NUC" |
|:---:|:---:|:---:|:---:|:---:|:---:|
| **Chainsaw model** | - | - | - | - | * |
| **Operator** | ** | ** | * | - | * |
| **Chainsaw model x Operator** | - | - | - | NA | - |

Signif. codes: '**', *p* < 0.01; '*', *p* < 0.05; '-', not significant; 'NA', no data.

Considering both chainsaw models and operators, the average gross daily productivity was 1.77 m$^3$h$^{-1}$, with a minimum of 1.42 m$^3$h$^{-1}$ (value found in BAT) and a maximum of 2.11 m$^3$h$^{-1}$ (value found in BAT). Furthermore, the overall average gross daily productivity of the two operators was analysed, and a significantly higher productivity was recorded for operator 2 (overall average daily gross productivity for operator 1 was 1.62 m$^3$h$^{-1}$: 1.47 m$^3$h$^{-1}$ for BAT, 1.76 m$^3$h$^{-1}$ for MIX; for operator 2, it was 1.93 m$^3$h$^{-1}$: 1.93 m$^3$h$^{-1}$ for BAT, 1.92 m$^3$h$^{-1}$ for MIX; Table 5). The comparison of the average gross daily productivity of BAT and MIX in operator 2 did not show any significant difference, while significantly higher productivities were found using MIX by operator 1 (Table 5).

**Table 5.** Overall productivity dataset and statistics for each operator (1 and 2) and chainsaw models (BAT-battery MSA 300; MIX-petrol MS 261 C-M).

| | Gross | | | | Net | | | | Productivity ($m^3h^{-1}$) Felling/Processing | | | | Using Chainsaw "UC" | | | | Not Using Chainsaw "NUC" | | | |
|---|---|---|---|---|---|---|---|---|---|---|---|---|---|---|---|---|---|---|---|---|
| **Operator** | 1 | | 2 | | 1 | | 2 | | 1 | | 2 | | 1 | | 2 | | 1 | | 2 | |
| Chainsaw model | BAT | MIX | BAT | MIX | BAT | MIX | BAT | MIX | BAT | MIX | BAT | MIX | BAT | MIX | BAT | MIX | BAT | MIX | BAT | MIX |
| **Mean** | 1.47 a | 1.76 b | 1.93 a | 1.92 a | 2.35 a | 2.64 a | 2.89 a | 3.00 a | 2.62 a | 2.89 a | 3.12 a | 3.17 a | 3.30 a | 3.46 a | 4.13 a | 3.69 a | 5.87 a | 7.46 a | 7.85 a | 10.42 a |
| **Min** | 1.42 | 1.71 | 1.69 | 1.78 | 2.15 | 2.54 | 2.48 | 2.84 | 2.31 | 2.75 | 2.66 | 2.99 | 3.08 | 3.25 | 3.45 | 3.36 | 5.5 | 6.45 | 7.25 | 8.36 |
| **Max** | 1.55 | 1.82 | 2.11 | 2.09 | 2.46 | 2.81 | 3.11 | 3.21 | 2.86 | 3.05 | 3.38 | 3.39 | 3.47 | 3.83 | 4.5 | 4.28 | 6.47 | 9.33 | 8.18 | 12.74 |
| **SD** | 0.07 | 0.06 | 0.22 | 0.16 | 0.17 | 0.14 | 0.35 | 0.18 | 0.28 | 0.15 | 0.4 | 0.2 | 0.2 | 0.32 | 0.59 | 0.51 | 0.52 | 1.62 | 0.52 | 2.2 |
| **Overall operator** | 1.62 a | | 1.93 b | | 2.50 a | | 2.95 b | | 2.76 a | | 3.14 b | | 3.38 a | | 3.91 a | | 6.67 a | | 9.13 b | |

Different lowercase letters highlight a significant difference in the row, between chainsaw per operator (*p*-value < 0.05).

The average net daily productivity for operator 1 was 2.35 $m^3h^{-1}$ for BAT, with a minimum value of 2.15 $m^3h^{-1}$ and a maximum of 2.46 $m^3h^{-1}$. The comparison of the daily net productivity of BAT and MIX models did not show any significant differences (Table 5). In fact, MIX was only 0.29 $m^3h^{-1}$ higher than BAT (Table 5). For operator 2, the average net daily productivity was 2.89 $m^3h^{-1}$ for BAT, with a minimum value of 2.48 $m^3h^{-1}$ and a maximum of 3.11 $m^3h^{-1}$. The comparison of the daily net productivity of BAT and MIX did not show any significant differences. In fact, MIX was only 0.11 $m^3h^{-1}$ higher than BAT (Table 5). The overall average net productivity of the two operators was statistically different (Table 5). Analysing the felling and processing productivity of operator 1, similar average values were recorded using MIX (2.89 $m^3h^{-1}$) and BAT (2.62 $m^3h^{-1}$). Similar results were also observed for operator 2, who achieved the following average productivities in the felling and processing phases: 3.12 $m^3h^{-1}$ using BAT and 3.17 $m^3h^{-1}$ using MIX. The overall average felling and processing productivity of the two operators was statistically different (Table 5).

Comparing the average UC productivity, no significant differences were found for either operator 1 and operator 2 in the use of BAT and MIX (Table 5). Using MIX, the average UC productivity was 3.57 $m^3h^{-1}$, with a minimum value of 3.25 $m^3h^{-1}$ and a maximum of 4.28 $m^3h^{-1}$. Using BAT, it was 3.72 $m^3h^{-1}$, with a minimum value of 3.08 $m^3h^{-1}$ and a maximum of 4.50 $m^3h^{-1}$ (Table 5). No significant differences were also found between the two operators in terms of overall average UC productivity. Analysing the NUC productivity, our results show significant differences in terms of overall average productivity between operators 1 and 2 (Table 5). Considering the two chainsaw models, no significant differences were recorded between BAT and MIX used by operator 1 and by operator 2 (Table 5). Instead, statistically significant differences were found when the average NUC productivity values calculated for the two operators using BAT were considered.

## 4. Discussion

The direct comparison of work productivities between the two chainsaws was possible thanks to the comparable technical specifications of the two chainsaw models as well as the characteristics of the cut and processed trees (Table 3). Furthermore, the two workers operated under the same conditions, in terms of both trees and work organisation. In this context, the statistical analysis shows that the differences in productivities are due to the operator and not to the type of chainsaw (Table 4). This is in line with previous studies. It is well known that differences in productivities can be notable among different operators, even if operating under the same conditions and with the same equipment [31–33]. Accordingly, significant differences were found in the overall average productivities of the two operators in all the investigated productivities, with the exception of the UC phase (Table 5). Furthermore, no statistical differences were recorded among the two chainsaw models in any of the productivities investigated during the clear-cut operations, except for the gross productivity for operator 1 (Table 5). This can be explained by the fact that, when calculating gross productivity, the working times taken into account do not depend on the model of chainsaw used, but on other factors strictly related to operators' behaviour during the operations. As cited above, it is known that, under the same working conditions, different operators usually achieve different productivities [31]. Moreover, the operator has a great influence on productivity in most types of forestry works, and of all the factors that influence time consumption, the most difficult to keep constant is the operator [31]. Given that the UC productivity is similar between operators, and including only the work phases in which the chainsaw is used, the results show that variability is not due to the chainsaw model.

It is also important to underline that the productivity of felling and processing is mostly affected by the DBH of the tree being felled and by the distance between harvested trees. As confirmed by many authors [34–36], the tree diameter and inter-tree distance are the most important influencing factors on the productivity of felling. In light of the above, the findings of this research should not be considered for trees with larger DBH and different stands, where technological differences can change the performance of chainsaws

and their related productivity. Nonetheless, our findings comply with previous studies analysing battery chainsaws; however, they are limited in that they only focused on the comparison of cutting efficiency in cross-cutting operations [4,23].

In summary, in a conifer clear-cut site with small-medium tree diameters, BAT and MIX are equivalent in terms of productivity. Despite this result, in practice, it is not possible to affirm that battery-powered chainsaws can be used in forestry on par with petrol-powered ones. In fact, as reported in the results, 8–9 charges are needed within a workday. Considering the logistics, it is not possible to bring 8–9 batteries to the forest due to their heavy weight (close to 20 kg in total) and hence low practicality [37]. Moreover, in considering costs, the price for a battery is around 500 EUR, which means a very high initial investment in comparison with a petrol-powered chainsaw. For these reasons, the possibility of charging batteries at the worksite seems to be the current greatest limitation to the use of battery-powered chainsaws in forestry. Other limitations that can retard the development of battery engines in comparison with traditional petrol ones are related to the resource availability for lithium-ion cell manufacturing [38,39] and the oil industry's profits despite the great attention to reducing emissions into the atmosphere. These are critical aspects for the continuing development of electric engines.

Finally, the results show that, despite the lower cutting efficiency of the battery-powered chainsaw [4], felling and processing productivity was similar to the petrol one. Thus, as battery technology is innovating quickly, it is likely to further evolve and address the current drawbacks of battery life. The use of battery-powered chainsaws also brings important advantages for both the environment and the workers, such as reduced gas emissions, noise and vibrations. For these reasons, it is important to invest in the research and development of this technology to increase the operational efficiency of batteries and to solve the problem of recharging, in order to ensure a safer and healthier future for forest operators.

## 5. Conclusions

The objective of this research was to investigate the productivity achieved by battery- and petrol-powered chainsaws in real forestry operations, such as a conifer clear-cut. The two models proved to be comparable in terms of productivity. Our results show that the research hypothesis was confirmed, i.e., professional battery chainsaws are suitable for small-scale forestry operations and especially when acoustic and/or gas pollution reductions are necessary. Professional battery chainsaws were also found to be appropriate for small-scale processing operations at the landing when a whole tree harvesting system is applied and in all those cases in which there are work limitations in terms of noise and vibrations exposure. Despite the good performance achieved by battery chainsaws, including their reliability and productivity, they are still far from being regularly used in forestry. In particular, our research highlights that the frequent need to recharge the battery is the only and the most important drawback. However, the research in this field must continue to evaluate the performance of these tools for their suitability for forestry operations as technology in this field is improving quickly, especially in terms of battery weight and duration.

For all these reasons, new research in this field should take into consideration the battery duration in relation to its cost and to the cost of recharging, while also considering the whole technical life of the machine. These aspects could be useful for comparing battery-powered and petrol-powered machines in future studies aimed at evaluating the amount of emissions produced and the life cycle assessment (LCA).

**Author Contributions:** Conceptualization, E.M. (Enrico Marchi), A.L. and F.F.; methodology, E.M. (Enrico Marchi), A.L., C.F. and F.N.; software, C.F., A.L., N.F. and E.M. (Elena Marra); validation, all authors; formal analysis, C.F., E.M. (Elena Marra), A.L., F.N. and E.M. (Enrico Marchi); investigation, all authors; resources, F.N. and E.M. (Enrico Marchi); data curation, all authors; writing—original draft preparation, F.N., A.L., R.P.; E.M. (Elena Marra) and E.M. (Enrico Marchi); writing—review and editing, A.L. and F.N.; supervision, F.N. and E.M. (Enrico Marchi); funding acquisition, F.N. and E.M. (Enrico Marchi). All authors have read and agreed to the published version of the manuscript.

**Funding:** The publication was made with the contribution of the researcher Cristiano Foderi with a research contract co-funded by the European Union—PON Research and Innovation 2014–2020 in accordance with Article 24, paragraph 3a), of Law No. 240 of December 30, 2010, as amended, and Ministerial Decree No. 1062 of August 10, 2021. A. Laschi thanks the University of Palermo for internal funds, project "FFR2021_Laschi_Andrea".

**Data Availability Statement:** The data presented in this study are available on request from the corresponding author. The data are not publicly available due to privacy.

**Acknowledgments:** The authors would like to express their gratitude to the Municipality of Modigliana (province of Forli-Cesena) and to Alessandro Liverani for providing the authorised worksite, to Andreas Stihl S.p.A. for providing the chainsaws used in this study and to the two young forestry students, Jessica Scriva and Antonio Colletti, for their contribution during data collection in the field.

**Conflicts of Interest:** The authors declare no conflict of interest.

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
