# Peer review of "Comparing the Productivity of the Latest Models of Li-Ion Battery and Petrol Chainsaws in a Conifer Clear-Cut Site"

_forests, doi:10.3390/f14030585_

Round 1
Reviewer 1 Report
Thank you very much for giving me the opportunity to evaluate this article. The purpose of the article is interesting and very practical.
As far as the content is concerned, the paper has a clearly stated and unambiguous aim, with clear and concise presentation of data. In addition, its results respond to the aim of the study. The Discussion section is relevant and adequate. References are related.
It has got an informative and clear title as well as a logical structure and good statement and language.
However, I would suggest the following corrections:
Table 4 is difficult to understand. What is meant by “-“ and “*” symbols? Give an explanation below the table 4.
In L195 and L197 “Table 2” should probably become “Table5”
Reviewer 2 Report
The paper is on an interesting topic. Some linguistic improvements are necessary as well as partial reworking of some parts of the text. For further consideration, the following points should be addressed:
Title: The use of the term “avant garde” is a little too far-fetched. In fact, a product that can be sold isn’t avant-garde rather modern, contemporary or similar. Also, the title as is, doesn’t reflect a very important aspect of the study – battery duration.
Introduction: Could you include more information, if available, on the battery duration improvements since the launching of the battery chainsaws?
L13: … the battery-power Stihl MSA 300 and the petrol-powered….
L15: Please reword “ The working ….was calculated”.
L16: Could you also include the average DBH that corresponds to a volume of 0.64 m3?
L18: Please reword “using chainsaw”
L20: What do you mean by “one”?
L47: Please add some refrences on this point.
L48: It would be better to refer to them as “occupational health” rather than “physical” problems.
L53: Please explain w/w
L68: Please insert (8 hours) right after “working day”
L69: Consider using a more appropriate introductory word/sentence.
L71: conditions
L75-76: These lines could be re-located in text or even abandoned.
L89: What exactly do you mean by “regular”?
L95: .. of the Montebello…
L95: six vs 6
L109: … indicated as BAT …, … as MIX…
How many chainsaw operators participated in the study? What was their experience in years?
Analysis of the workday:
“Using the chainsaw” and “not using the chainsaw” wording approach is a little complicated. Please consider instead of that to distinguish between e.g Felling cut (presupposing that chainsaw is necessary) and Felling cut rest for all other non-chainsaw connected work elements. Consider doing necessary changes on this point.
Problems and drawbacks. Do you mean what is more commonly referred to as delays?
Table 2: In fact there is a “using chainsaw” work phase (Approach and tree study - using) that it is not included in the “using chainsaw” productivity.
L196: .. and the effective working hours per day.
L198: What is “felling and processing one”?
L209-201: …implementing a one-way ANOVA and a post-hoc Tukey's honestly 209 significant differences (HSD) test… to implementing one-way ANOVA and Tukey's honestly significant differences (HSD) post-hoc tests.
Table 3: Please consider changing row order e.g “minimum” next to “maximum” and “mean” next to “SD’. The authors should also make use of superscript letters for demoting the statistical differences or lack of
P233: per battery charge…
Table 4 should be changed by including productivity values. What is the meaning of using single and double stars? All related information must be included in the Table captions or footnote.
Figure 1: It can be removed, as all information included is also available at Table 5. Furthermore, the scales used in Figure are not helping the reader (e.g. in Gross productivity the max y-axis value could be 5 m3 h-1 so that differences are more easiliy identified).
L304: … as well as the attributes of the…
L337: within a (or per) workday
L354: Please consider changing the introductory “Here”.
Reviewer 3 Report
This manuscript is well structured and well written, which is easy to follow. The figures and tables are neat and easy to understand. The methodology is thoroughly explained and the work overall seems to be skillfully performed. Generally, I think this manuscript can be accepted for publication after minor revision.
1. Please avoid using the first personï¼›
2. A little bit of rearranging of the abstract is needed maybe. Abstract is usually divided into four parts WHY, WHAT, HOW, and main conclusions.
WHY This section usually contains one or two lines mainly defining what is the objective of the study or this work was done.
WHAT This is the main portion of the abstract. It contains what was done. Like what simulations have been performed what kind of parametric studies are done to support the WHY section.
HOW In this section you will define how u have achieved the WHAT section points. What kind of methodologies you have utilized to achieve the goals defined in WHAT section?
3. Provide more information for the cutting power and efficiency in the introduction, such as "Energy Efficiency Optimization for Machining of Wood Plastic Composite"
4. Flow chart can be supplemented for the studied workday and the tree felling and processing cycle.
5. he did not use it? it is hard to follow.
6. Supplement the limitation of your work.
7. Quote others as the explanation of the phenomenon, and supplement the references.
Round 2
Reviewer 2 Report
The authors have tried to address the reviewer's comments.
Nevertheless, a linguistic check is still necessary in many parts of the text.